# Genome Sequencing and Comparative Genomic Analysis of Attenuated Strain *Gibellulopsis nigrescens* GnVn.1 Causing Mild Wilt in Sunflower

**DOI:** 10.3390/jof10120838

**Published:** 2024-12-04

**Authors:** Baozhu Dong, Wanyou Liu, Yingjie Zhao, Wei Quan, Lijun Hao, Dong Wang, Hongyou Zhou, Mingmin Zhao, Jianxiu Hao

**Affiliations:** 1College of Horticulture and Plant Protection, Inner Mongolia Agricultural University, Hohhot 010010, China; dongbaozhu2020@imau.edu.cn (B.D.); yingjie.zhao@syngentagroup.cn (Y.Z.); quanwei9810@163.com (W.Q.); hlj2018nongxue@126.com (L.H.); wangdong@imau.edu.cn (D.W.); mingminzh@163.com (M.Z.); 2Key Laboratory of Biological Pesticide Creation and Resource Utilization, Education Department of Inner Mongolia, Hohhot 010011, China; 3Grassland Research Center, Chinese Academy of Forestry, Beijing 100091, China; m15548290183@163.com

**Keywords:** *Gibellulopsis nigrescens*, genome sequencing, gene prediction, annotation, comparative genomics

## Abstract

*Gibellulopsis nigrescens*, previously classified in the *Verticillium* genus until 2007, is an attenuated pathogen known to provide cross-protection against *Verticillium* wilt in various crops. To investigate the potential mechanisms underlying its reduced virulence, we conducted genome sequencing, annotation, and a comparative genome analysis of *G. nigrescens* GnVn.1 (GnVn.1), an attenuated strain isolated from sunflower. The genome sequencing and annotation results revealed that the GnVn.1 genome consists of 22 contigs, with a total size of 31.79 Mb. We predicted 10,876 genes, resulting in a gene density of 342 genes per Mb. The pathogenicity gene prediction results indicated 1733 high-confidence pathogenicity factors (HCPFs), 895 carbohydrate-active enzymes (CAZys), and 359 effectors. Moreover, we predicted 40 secondary metabolite clusters (SMCs). The comparative genome analysis indicated that GnVn.1 contains more CAZys, SMCs, predicted effectors, and HCPF genes than *Verticillium dahliae* (VdLs.17) and *Verticillium alfalfae* (VaMas.102). The core–pan analysis results showed that GnVn.1 had more specific HCPFs, effectors, CAZys, and secreted protein (SP) genes, and lost many critical pathogenic genes compared to VdLs.17 and VaMs.102. Our results indicate that the GnVn.1 genome harbors more pathogenicity-related genes than the VdLs.17 and VaMs.102 genomes. These abundant genes may play critical roles in regulating virulence. The loss of critical pathogenic genes causes weak virulence and confers biocontrol strategies to GnVn.1.

## 1. Introduction

*Gibellulopsis nigrescens* was initially identified as a soil-borne saprophytic fungus [1,2]. Although it was discovered a long time ago, there are no available data on its genome. Zhou et al. isolated *G. nigrescens* from wilting sugar beet plants [3]. Hu also obtained a *G. nigrescens* strain from alfalfa [4]. These isolates from sugar beet and alfalfa both caused wilt symptoms [3,4].

It has also been proven that *G. nigrescens* can infect many crops, including potato, cotton, sunflower, and soybean, causing weak wilt and slight vascular browning [5,6,7,8]. Therefore, isolates of this species are considered attenuated vascular pathogens, saprophytic fungi, or endophytes [9,10]. Because of the attenuated virulence of *G. nigrescens*, it is used in many crops to provide cross-protection against *Verticillium* wilt, a soil-borne vascular disease that occurs worldwide and leads to heavy losses in plant farming [11,12]. In a previous study, the pre-inoculation of *G. nigrescens* from peppermint significantly reduced wilt severity 7 and 9 days after challenge inoculation [13]. In addition, GnVn.1, an attenuated isolate from sunflower, was used to control the *Verticillium* wilt of sunflower and potato in a greenhouse [5,14]. Zhu et al. [15] and Vagelas [16] both isolated attenuated *G. nigrescens* strains from cotton, and they were found to cross-protect cotton against highly virulent strains of *V. dahliae* species. Subsequently, a study by Feng et al. showed that the pre-inoculation of *G. nigrescens* CEF08111 upregulated 710 resistance-related genes, indicating that *G. nigrescens* CEF08111 induces immunity in cotton [6]. Therefore, *G. nigrescens* is a potential biocontrol resource for *Verticillium* wilt management.

*G. nigrescens* was first isolated from potato, and it was named *Verticillium nigrescens* because its morphological characteristics are similar to those of species of the *Verticillium* genus [17]. It produces circular or elliptical transparent single-cell conidia. The phylogenic analysis results based on LSU and ITS sequences show that *V. nigrescens* is not congeneric with *V. dahliae* or *V. albo-artum*, two common species of the *Verticillium* genus. *Gibellulopsis*, which has a close genetic relationship with *Verticillium*, is a more suitable genus for *V. nigrescens* [18]. *G. nigrescens* produces chlamydospores, which are resistant to environmental stress, instead of microsclerotia or resting mycelium, as produced by *Verticillium* species [19]. As a result, it can colonize soil or plants for a long time [20].

Currently, there are no genomic data available for *G. nigrescens*. A comparative genome analysis of *G. nigrescens* and species of the *Verticillium* genus has not been reported. Thus, the genetic relationship between *G. nigrescens* and *Verticillium* species is not clear. In this study, we performed genome sequencing, annotation, and a comparative genome analysis to identify pathogenic genes and establish a basis for future hypovirulence and biocontrol mechanism studies.

## 2. Materials and Methods

### 2.1. Fungal Growth and Genomic DNA Extraction

For use in genomic DNA extraction, we cultured the mycelium of GnVn.1 on PDA medium covered with a layer of glass paper. The plates were incubated in the dark at 25 °C for 14 days. Afterward, we scraped the mycelium off with a scalpel and ground it into a powder using liquid nitrogen. We then extracted the genomic DNA using the CTAB method [21].

### 2.2. Genome Sequencing and Assembly

For sequencing on the PacBio platform, we constructed a library using a SMRT Bell™ Template Kit (PacBio, Menlo Park, CA, USA). We randomly sheared high-quality genomic DNA (20 μg) into fragments of approximately 20 kb using a Covaris g-TUBE, and then we performed end repair according to the kit instructions. We then ligated the DNA fragments to a hairpin structure and purified them with AMPure PB to generate a SMRT Bell template. We assessed the library quality using a Qubit^®^ 2.0 Fluorometer (Thermo Scientific, Waltham, MA, USA) and measured the insert fragment size with an Agilent 2100 (Agilent Technologies, Santa Clara, CA, USA). We then sequenced the SMRT Bell library using the PacBio Sequel IIe platform (Novogene, Beijing, China).

For sequencing on the Illumina platform, we sheared the genomic DNA to approximately 350 bp using the ultrasonic disintegration method (Covaris, Woburn, MA, USA). We constructed a library with an NEBNext^®^ Ultra™ DNA Library Prep Kit (NEB, Ipswich, MA, USA) for Illumina, followed by quantification and quality assessment using a Qubit 2.0 and an Agilent 2100, respectively. We then sequenced the library on the Illumina PE150 platform (Illumina, Inc., San Diego, CA, USA).

We filtered low-quality reads (less than 500 bp) using SMRT Link v5.0.1 software to generate clean reads (https://www.pacb.com/smrt-link/ (accessed on 3 May 2024)). We performed de novo genome assembly with the clean reads, utilizing the automatic error correction function of the SMRT portal [22,23]. We selected long reads (>6000 bp) as the seed sequence and aligned the shorter reads to this seed sequence using BLAST (Version 2.12.0) to further improve its accuracy. The variant caller module of SMRT Link software was employed with the arrow algorithm to correct and count the variant sites in the preliminary assembly results. Finally, we evaluated the completeness of the assembly using Benchmarking Universal Single-Copy Orthologs (BUSCO, v5.8.1) against fungi_odb10 [24].

### 2.3. Genome Component Prediction

The gene components analyzed included coding genes, repetitive sequences, and non-coding RNA. We used both ab initio and homology-based methods for coding gene prediction. For the ab initio prediction, we employed the Augustus 2.7 program with default settings. For the homology-based prediction, we used GeneWise 2.4.1 and the uniref90 non-redundant protein database, comparing the results with two related species, *Verticillium dahliae* and *Verticillium albo-atrum*. We predicted interspersed repetitive sequences using RepeatMasker (Version open-4.0.5) [25]. Tandem repeats were identified using Tandem Repeats Finder (Version 4.07b). Transfer RNA (tRNA) genes were predicted with tRNAscan-SE [26], while ribosomal RNA (rRNA) genes were identified using rRNAmmer [27]. Finally, we predicted small RNA (sRNA), small nuclear RNA (snRNA), and microRNA (miRNA) by performing a BLAST analysis against the Rfam database [28].

### 2.4. Gene Function Prediction and Annotations

We predicted gene functions using seven databases: GO, KEGG, Cluster of Orthologous Groups (KOG), Non-Redundant (NR) Protein Database, Transporter Classification Database (TCDB), P450, and Swiss-Prot [29]. We performed a whole-genome BLAST search against these databases, identifying homologous genes with an E-value less than 10^−5^ and a minimum alignment length greater than 40%. We predicted SPs using the SignalP 6.0 online tool (https://services.healthtech.dtu.dk/services/SignalP-6.0/ (accessed on 12 October 2024)). SMCs were identified by searching the antiSMASH fungal version (https://fungismash.secondarymetabolites.org/#!/start (accessed on 20 July 2024)) [30]. For HCPFs and a drug resistance analysis, we used the Database of Fungal Virulence Factors and the Pathogen–Host Interaction Database (PHI-base: https://doi.org/10.1093/nar/gkab1037 (accessed on 20 July 2024)) [31]. We predicted CAZys using the CAZy database (http://www.cazy.org/ (accessed on 21 July 2024)) [32].

### 2.5. Comparative Genomics Analysis

*G. nigrescens* is regarded as an attenuated pathogen affecting various crops, and it was classified in the *Verticillium* genus until a report by Zare et al. [18]. Species of the *Verticillium* genus are highly virulent on many plants. Additionally, *G. nigrescens* and species of the *Verticillium* genus all cause wilt symptoms, indicating a similar pathogenic mechanism among these pathogens. To compare the pathogenicity-related genes between *G. nigrescens* and species of the *Verticillium* genus, we conducted a core and specific gene analysis (core–pan analysis) and constructed a phylogenetic tree based on this analysis [33]. We utilized genomic data from two *Verticillium* species, namely, *V. dahliae* VdLs.17 and *V. alfalfae* VaMs.102, for our core–pan analysis. We analyzed core and specific genes using CD-HIT software (Version 4.6.1), which rapidly clusters similar proteins with thresholds of a 50% pairwise identity and a 0.7 length difference cutoff in amino acids. We then created a Venn diagram to illustrate the relationships among the species. Finally, we searched the specific genes against the PHI-base, EffectorP database, and CAZy database to predict pathogenicity-related genes.

### 2.6. Phylogenic Tree Construction

We performed a phylogenetic analysis to examine the genetic relationship between the attenuated strain GnVn.1 and two virulent strains of the *Verticillium* genus. Species of the *Fusarium* genus also cause wilt symptoms. The two strains *Fusarium oxysporum* Fo47 and *F. solani* Fusso1 are both nonpathogenic on the host. For comparison, we included the genomic data of *F. oxysporum* Fo47 and *F. solani* Fusso1 as outgroups. We constructed a phylogenetic tree using PhyML3.1 software, applying the maximum likelihood method with 1000 bootstrap replicates for support [34].

## 3. Results

### 3.1. Genome Assembly and General Characteristics

The procedure of genome sequencing, annotation and comparative analysis are showed in Figure 1. After filtering out low-quality reads, we generated a total of 1,564,753 high-quality reads from the PacBio sequencing platform, covering 7,724,321,021 base pairs (bps) in total, with a mean length of 4936 bps and an N50 read length of 6636 bps. The genome assembly results revealed that the genome size of GnVn.1 is 31.79 Mb. This genome comprises 22 curated contigs, with a maximum contig length of 6,280,609 bps and an N50 contig length of 2,979,356 bps. The completeness of the assembly, assessed using BUSCOs, was found to be 98.3%. We predicted 10,876 coding genes in the genome, resulting in a gene density of 342 genes per Mb. Additionally, we identified 188 tRNA coding genes. Our analysis revealed a total of 640 repetitive sequences, including LTRs, DNAs, LINEs, SINEs, RCs, and other unknown transposons, accounting for 0.24% of the genome and totaling 79,953 bps (Table 1). We submitted the genome data to the NCBI database, where it is accessible under the accession number SAMN43317469.

### 3.2. Gene Annotation

To predict the biological functions of the genes, we searched the predicted genes against seven databases: GO, KEGG, CKOG, NR, TTCDB, P450, and Swiss-Prot. As a result, 93.41% (10,159) of the genes in GnVn.1 were successfully annotated using these seven protein databases. Additionally, we identified 188 tRNA genes, 58 rRNA genes, and 640 repetitive sequences in the GnVn.1 genome.

### 3.3. KEGG and GO Enrichment

A total of 3784 genes were annotated using the KEGG database. We performed a KEGG enrichment analysis based on these annotated genes. The KEGG enrichment results indicated that 523 genes were enriched in five Cellular Processing pathways, 264 genes in three Environmental Information Processing pathways, 782 genes in the Genetic Information Processing pathway, 776 genes in twelve Human Disease pathways, 2956 genes in twelve Metabolism pathways, and 602 genes in ten Organismal System pathways (Figure 2). The GO analysis results showed significant enrichment in 11 Molecular Functions, 17 Cellular Components, and 23 Biological Processes (Figure 3).

### 3.4. Prediction of Pathogenicity-Related Genes

Previous studies have indicated that the proteins in plant pathogens involved in HCPFs, secondary metabolite (SM) synthesis, CAZys, and secretory proteins (SPs) significantly affect the virulence of phytopathogens. Therefore, we searched for pathogenicity-related genes in GnVn.1 against several databases, including PHI-base, the CAZy database, EffectorP, antiSMASH, SignalP 6.0, and TMHMM 2.0.

#### 3.4.1. Prediction of HCPFs

Using the PHI-base, we predicted a total of 1733 genes as HCPFs. Among these, 682 genes were annotated as having no effect on pathogenicity, while 647 genes were associated with reduced virulence. Additionally, 135 genes were annotated as resulting in a loss of pathogenicity, 88 genes as lethal, and 28 genes as associated with increased virulence (hypervirulence). Moreover, two genes were annotated as enhancing antagonism, twenty-four genes as effectors (plant avirulence determinants, AVR), and four genes as chemical targets (resistant to chemicals). Furthermore, 123 HCPF entries remained unannotated (NA) (Figure 4A).

#### 3.4.2. CAZys

Pathogenic fungi can degrade plant cell walls to facilitate the penetration of hyphae and the absorption of nutrients by secreting CAZys [35]. We annotated 895 genes in the CAZy database, which belong to the following families: carbohydrate-binding modules (CBMs), carbohydrate esterases (CEs), glycoside hydrolases (GHs), glycosyl transferases (GTs), polysaccharide lyases (PLs), and auxiliary activities (AAs). Specifically, among these predicted CAZys, 123 genes were annotated as CBMs, 69 as CEs, 422 as GHs, 36 as PLs, and 106 as AAs (Figure 4A).

#### 3.4.3. Secretome and Effector

We predicted SPs using the SignalP and TMHMM tools. Our screening results identified 1082 proteins as SPs. Secretory CAZys play a crucial role in virulence and nutrient uptake for pathogenic fungi [36,37,38,39]. Among the predicted secretory protein candidates, we identified 444 genes as CAZys, which included 181 GHs, 39 CEs, 74 AAs, 35 PLs, and 11 GTs. Additionally, we found 13 SPs that contained a CBM domain (Appendix A).

Plant pathogenic fungi typically secrete apoplastic and cytoplasmic effectors during plant–pathogen interactions to facilitate immunity recognition and regulate host immunity [40]. We utilized the EffectorP 3.0 online tool to predict which SPs are potential effectors. As a result, we identified 359 effector candidates: 125 apoplastic effectors, 126 cytoplasmic effectors, and 108 effectors that are both apoplastic and cytoplasmic (Figure 4A). By comparing the putative effectors with the PHI-base and CAZy database, we annotated seventy-two genes as CAZys (twenty-two AAs, eleven CEs, nineteen GHs, nineteen PLs, and three CAZys with CBM domains) and fifty genes as HCPFs (twelve plant avirulence determinants, three increased virulence factors, one loss-of-virulence factor, sixteen reduced virulence factors, sixteen factors with unaffected pathogenicity, and two unannotated) (Appendix A).

#### 3.4.4. SMs

Previous studies have demonstrated that the SMs produced by plant pathogens play a significant role in the interaction between plants and pathogens. SMs contribute to the formation of infection-related structures and resting mycelium, serve as fungal toxins, and regulate plant resistance [41,42]. We predicted that 40 gene clusters, comprising 556 genes, were associated with SMs. Among these clusters, we identified five as non-ribosomal peptide synthase (NRPS) and six as fungal ribosomally synthesized and post-translationally modified peptides (RiPPs). Additionally, we identified eight clusters as type 1 polyketide synthase (T1PKS) and one cluster as type 3 polyketide synthase (T3PKS). Furthermore, we predicted six clusters to be terpene biosynthesis-related and one cluster to be phosphonate-related. We also identified three clusters as NRPS/T1PKS, T1PKS/indole, and T1PKS/T3PKS, respectively (Figure 4B).

### 3.5. Comparative Genomic Analysis

Previously, *G. nigrescens* was classified as a member of the *Verticillium* genus, which causes vascular wilt in multiple plants. Infection by *G. nigrescens* results in the mild wilting of cotton, sunflower, peppermint, and potato [5,8,14,15]. This pathogen has the potential to provide cross-protection against virulent *Verticillium* strains in plants [13]. To investigate its phylogenetic relationships, we constructed a phylogenetic tree for GnVn.1, VdLs.17, and VaMs.102 based on the core–pan analysis results. We used a nonpathogenic strain, *Fusarium oxysporum* Fo47, and an endophytic strain from *Arabidopsis thaliana*, *F. solani* Fusso1, as outgroups. The phylogenetic analysis revealed that VdLs.17 and VaMs.102 clustered together in a clade, with a support rate of 100%. In contrast, GnVn.1 served as an outgroup in the *Verticillium* genus (Figure 5).

SMCs, HCPF, CAZys, effectors, and SPs significantly contribute to the virulence of plant pathogens. Therefore, we compared these four types of genes across different species. Our results show that GnVn.1 possesses more SMCs (40) than VdLs.17 (32 SMCs) and VaMs.102 (30 SMCs). The SMCs of phosphonate and indole were only present in the GnVn.1 genome. The SMCs of isocyanide-nrp were only present in the VdLs.17 and VaMs.102 genomes (Appendix A). Similarly, GnVn.1 exhibited a greater number of predicted HCPF genes, effectors, CAZys, and SP genes than VdLs.17 and VaMs.102. Additionally, we found that two nonpathogenic *Fusarium* species had more pathogenic genes than the two pathogenic strains of the *Verticillium* genus. These findings suggest that the abundance of SMCs, HCPFs, CAZys, and SPs may contribute to the weaker virulence of GnVn.1 (Figure 5).

We subsequently compared the CAZys and secreted CAZys of GnVn.1, VdLs.17, and VaMs.102. The results indicate that GnVn.1 contains more CAZys and secreted CAZys in its genome. Specifically, GnVn.1 exhibits a higher abundance of secreted CBMs, GTs, and AAs than VdLs.17 and VaMs.102 (Figure 6A and Appendix A).

### 3.6. Analysis of Specific and Dispensable Gene Groups

The core–pan analysis results show that the virulent strain VdLs.17 has 2438 specific gene groups, VaMs.102 has 3208 specific gene groups, and the hypovirulent strain GnVn.1 has 4862 specific gene groups (Figure 6B). GnVn.1 contains more specific gene groups than both VdLs.17 and VaMs.102. We annotated the specific genes of these different strains. The GnVn.1 genome includes 439 specific HCPFs and 231 specific effectors. In contrast, VdLs.17 contains only 171 specific HCPFs and 54 effectors, while VaMs.102 has 258 specific HCPFs and 78 specific effectors (Figure 6C,D). The specific effectors and HCPFs in GnVn.1 outnumber those in both strains of the *Verticillium* genus. Among the specific HCPFs, we identified thirteen genes as effectors: ten lysin motif (LysM) genes and three CDIP4 genes. In comparison, VdLs.17 and VaMs.102 contain only two and four specific effectors, respectively. Compared with VdLs.17 and VaMs.102, GnVn.1 has four down-regulating virulence HCPFs, of which gene silencing or mutation transformants showed increased virulence (Appendix A). These results indicate that GnVn.1 has a rich abundance of specific predicted effectors and HCPFs, which may be related to its attenuated virulence.

We also analyzed the dispensable HCPFs of three stains and predicted 96 critical HCPFs of VdLs.17 and VaMs.102, which were absent in GnVn.1. Among these HCPFs, four contribute to the virulence of *V. dahliae* or *V. longisporum*; twenty-nine are homologous with the pathogenic genes of *Magnaporthe oryzae*; twenty-seven are homologous with the pathogenic genes of species of the *Fusarium* genus; and eleven are homologous with the pathogenic genes of plant pathogens, such as *Leptosphaeria maculans*, *Bipolaris maydis*, *Botrytis cinerea*, *Colletotrichum acutatum*, and *Colletotrichum gloeosporioides* (Appendix A). The results indicate that the absence of critical pathogenic genes leads to weak virulence and confers biocontrol function to GnVn.1.

## 4. Discussion

*G. nigrescens* is a hypovirulent pathogen that causes mild wilting and contributes to cross-protection against *Verticillium* wilt in many plants. In this study, we performed genomic sequencing and annotated the gene functions of *G. nigrescens* isolated from sunflower. Additionally, we compared the genomes of GnVn.1, VdLs.17, and VaMs.102. Our results indicate that the GnVn.1 genome contains more predicted SMCs, specific HCPFs, and effector genes than the VdLs.17 and VaMs.102 genomes. Among the specific HCPFs, we identified thirteen effectors, which is significantly higher than that found in VdLs.17 (4) and VaMs.102 (5).

Several strains of *G. nigrescens* have been identified as either attenuated pathogens or endophytes in different plants. *G. nigrescens* strains provide cross-protection against Verticillium wilt in various crops. GnVn.1, isolated from sunflower, can protect potato and sunflower against virulent strains and reduce the level of wilting [5,14]. The strain *G. nigrescens* CEF08111, isolated from cotton, also protects cotton against *Verticillium* wilt. Studies have indicated that the SMs of CEF08111 influence mycelium growth, yield, and conidia shape [15]. Additionally, *G. nigrescens* CEF08111 can induce multiple defense-related pathways in cotton [6]. Moreover, mycotoxins such as ochratoxin A from *Aspergillus* and various mycotoxins from the *Fusarium* genus can even induce hypersensitive reactions in plant cells [41,43]. Previous studies have demonstrated that SMs play important roles in eliciting plant resistance [42,44,45]. In this study, we identified more SMCs in GnVn.1 than in VdLs.17 and VaMs.102. These results suggest that the large number of SMCs in *G. nigrescens* provides potential functions for inducing plant resistance.

CAZys play critical roles in plant pathology by contributing to fungal growth and virulence [38]. Extracellular CAZys degrade cell walls, facilitating fungal pathogen invasion [46]. Additionally, the degradation products of CAZys can act as damage-associated molecular patterns (DAMPs), which elicit plant resistance and play important roles in basal immunity recognition [47]. Many CAZys in *V. dahliae* have also been shown to contribute to its virulence [48,49]. The results obtained by Yan et al. indicated that a weakly virulent strain of Sclerotium rolfsii had fewer secreted CAZys than a highly virulent strain [50]. In contrast, our results show that GnVn.1 contains more CAZys and secreted CAZys in its genome. Generally, saprotrophic fungi possess more CAZys than plant fungal pathogens. However, necrotrophic fungal pathogens tend to be more abundant in CAZys [51]. These findings suggest that *G. nigrescens* functions as a weakly parasitic or saprotrophic fungus in soil, which explains its low virulence in many crops.

*G. nigrescens* is typically isolated from soil and plants exhibiting mild wilting symptoms, including potato, cotton, and sunflower [5,52]. Consequently, strains of this species have been identified as attenuated pathogens or endophytes. GnVn.1 has demonstrated weak virulence on potato and sunflower. In contrast, Zhou et al. showed that *G. nigrescens* could infect sugar beet, causing typical wilting symptoms [3]. We identified GnVn.1 as having a greater abundance of various pathogenic genes in its genome. Previous studies have indicated that *G. nigrescens* is a soil-borne fungus with a wide host range and exhibits various degrees of virulence toward different plants. Pathogenic strains of the *Verticillium* genus can cross-infect multiple hosts and exhibit different degrees of virulence on different plants [53]. These results suggest that the differentiation of pathogenicity in plant pathogens relates to the variety and quantity of pathogenic genes present in their genomes. Further investigation is needed to understand the virulence of GnVn.1 on sugar beet and the pathogenicity differentiation among *G. nigrescens* strains.

Plant biotrophic and hemi-biotrophic pathogens secrete PAMPs and effectors to regulate plant immunity [40]. Avirulent genes (*AVRs*) in plant pathogens code for subcellular effectors that induce plant immunity [54]. The specific interaction between *AVR* and resistant genes (R), known as the gene-for-gene system, triggers strong plant immunity, resulting in an abundant ROS burst and, in some cases, programmed cell death [55,56]. To date, many *AVR-R* pairs have been identified, including *AVR-Pita*, *Ave1-Ve1*, and *Rds-Rih* [57,58,59,60]. Therefore, AVR proteins act as specific elicitors of plant resistance by recognizing R proteins, which leads to a hypersensitive reaction. In our study, we identified a greater number of predicted specific *AVRs* in GnVn.1 (thirteen) than in VdLs.17 (two) and VaMs.102 (four) based on core–pan analysis results. These findings suggest that the abundance of specific *AVR*s may enhance host immunity, contributing to the hypervirulent phenotype of GnVn.1. The function of these specific *AVRs* warrants further investigation.

Among the predicted specific HCPF groups of three species, we identified ten genes annotated as LysM effectors in the genome of GnVn.1. In contrast, we predicted only one LysM effector in the specific HCPF group of VdLs.17 and two in VaMs.102. LysM effectors are widely present in the genomes of fungi and bacteria [61,62] and help prevent chitin-triggered immunity in plants by competitively binding to chitin oligosaccharides [63]. A previous study also showed that LysM proteins are involved in the appressorial function of *Colletotrichum higginsianum* [64]. Interestingly, the core LysM effectors of *V. dahliae* do not contribute to virulence in the infection process. However, Vd2LysM, an additional lineage-specific effector of VdLs.17, enhances virulence on tomato. We speculated that the secreted LysM proteins of GnVn.1 contributed to reduced virulence and cross-protection. The function of the specific effector genes in GnVn.1 requires further investigation.

*G. nigrescens* usually shows weak virulence on multiple plants. Through a dispensable gene analysis, we discovered that many pathogenic genes (*VdMsb*, *VdMsn2*, *VdQase*, and *Vta2*) were absent from the *G. nigrescens* genome. VdMsb, a transmembrane mucin, is located upstream of the MAPK signaling pathway and is necessary for full virulence [65]. VdMsn2 is a C_2_H_2_ transcription factor, contributing virulence by regulating hyphal growth [66]. VdQase confers *V. dahliae* virulence by counteracting host defense [67]. Vta2 is highly conserved in filamentous fungi, required for the systemic infection and microsclerotia formation of *V. longisporum* [68]. In addition, we also predicted many pathogenic genes in the VdLs.17 and VaMs.102 genomes, which were homologous to other plant pathogens. These genes are absent from the GnVn.1 genome. These results indicate that the loss of these genes leads to weak virulence and confers cross-protection to GnVn.1.

We identified three cell death-inducing protein (CDIP) genes in the specific HCPF group of GnVn.1, while VdLs.17 and VaMs.102 each contained one specific CDIP gene. Five CDIPs in *Magnaporthe oryzae* (MoCDIP1 to 5) can induce host cell death and serve as innate immunity elicitors [69]. MoCDIP4 binds to OsDjA9, a member of the heat shock–dynamin protein complex, in order to alter mitochondrial fission and reduce plant immunity [70]. *G. nigrescens* typically exhibits hypovirulence in plants. We speculate that the higher number of elicitors may induce robust innate immunity, preventing effectors from overcoming the hypersensitive response, which results in decreased pathogenicity.

*G. nigrescens* typically exhibits hypovirulence in plants. We predicted specific immunity elicitors and effectors in the *G. nigrescens* genome. The mechanisms by which elicitors regulate plant immunity need further study. We also discovered a loss of pathogenic genes in the GnVn.1 genome compared with in VdLs.17 and VaMs.102. Further studies are needed to determine whether deletion mutants of the pathogenic genes of virulent strains could be applied to *Verticillium* wilt management. Moreover, the safety of plant disease control using attenuated strains needs further evaluation.

## 5. Conclusions

In this study, we performed genome sequencing and a comparative genome analysis on *G. nigrescens*, an attenuated pathogen known to provide cross-protection against *Verticillium* wilt in various crops. The genome size was found to be 31.79 Mb and encoded 10,876 genes, with a gene density of 342/Mb. GnVn.1 also had more specific pathogenicity-related genes than VdLs.17 and VaMs.102. GnVn.1 exhibited a loss of many critical pathogenic genes compared with VdLs.17 and VaMs.102. We speculated that the abundance of pathogenicity-related genes regulates virulence, and that the loss of critical pathogenic genes leads to weak virulence. These results indicate that *G. nigrescens* could be used for *Verticillium* wilt management following a safety evaluation in the field. Many specific pathogenicity-related proteins, especially immunity elicitors, could be developed as agents inducing plant immunity against pathogen invasion.

## Figures and Tables

**Figure 1 jof-10-00838-f001:**
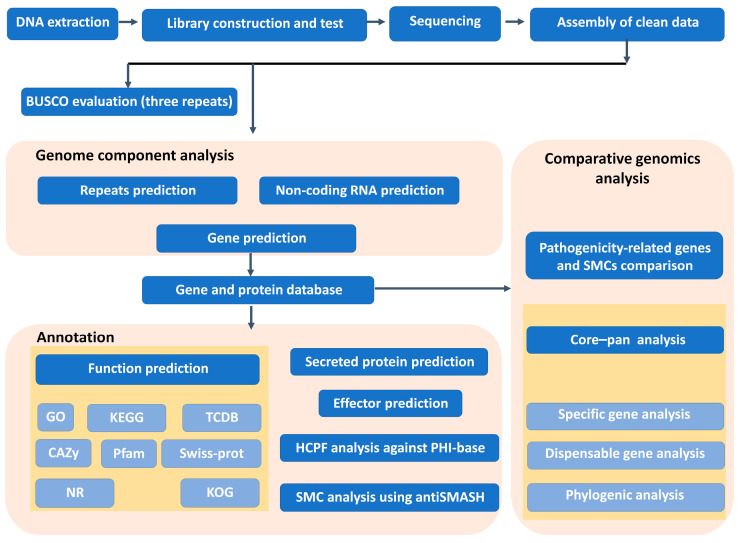
Steps in genome sequencing, annotation, and comparative genomics analysis.

**Figure 2 jof-10-00838-f002:**
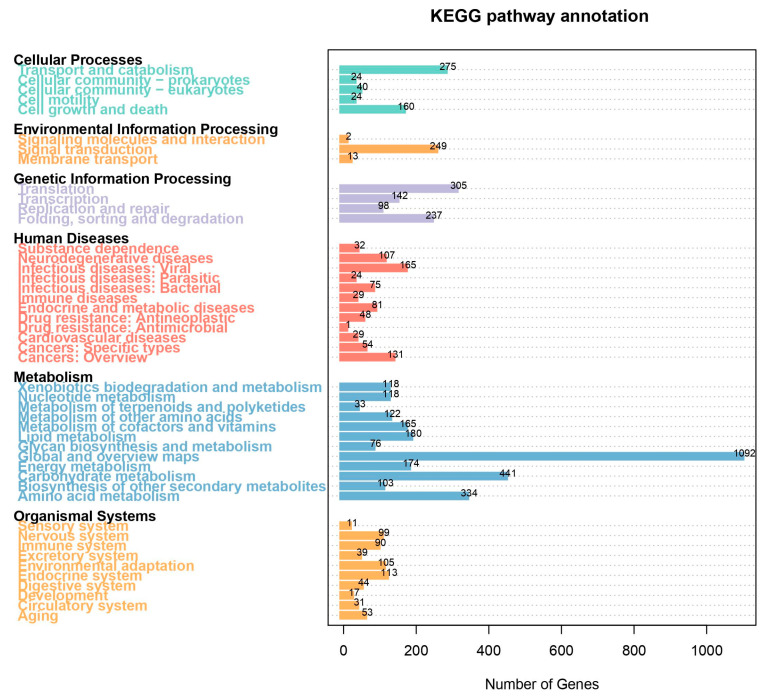
KEGG enrichment results.

**Figure 3 jof-10-00838-f003:**
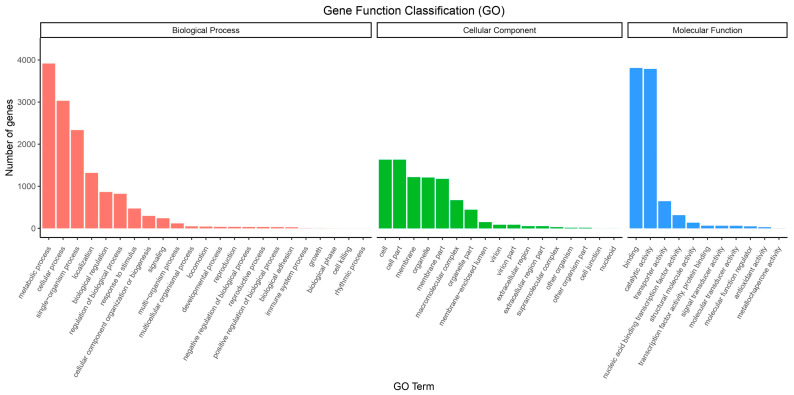
GO enrichment results.

**Figure 4 jof-10-00838-f004:**
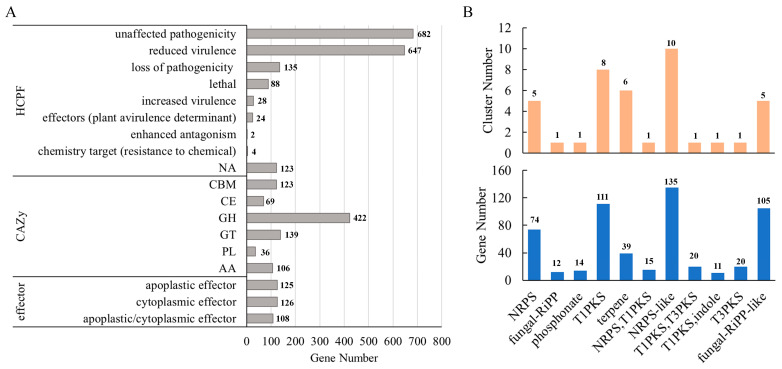
Prediction results of pathogenicity-related genes. (**A**) Number of predicted high-confidence pathogenicity factor (HCPF), carbohydrate-active enzyme (CAZy), and effector genes. (**B**) Number of predicted secondary metabolite clusters (SMCs) and genes. The column and label represent the number of genes and clusters.

**Figure 5 jof-10-00838-f005:**
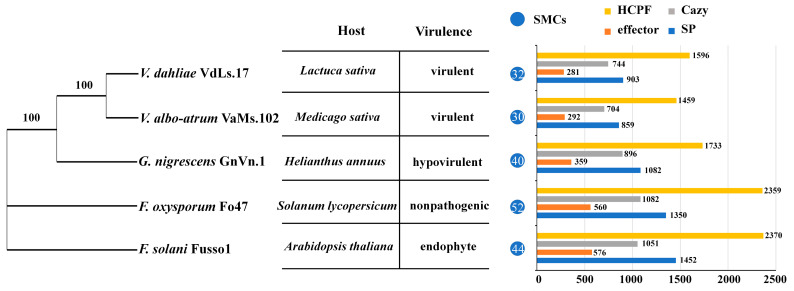
Comparative genome results of GnVn.1, VdLS.17, and VaMs.102. Phylogenetic tree was constructed using PhyML software, applying the maximum likelihood method with 1000 bootstrap replicates (**left**). The number represents the supporting rate. The table shows the host and virulence of fungi (**middle**). The column and label represent the number of genes and clusters (**right**).

**Figure 6 jof-10-00838-f006:**
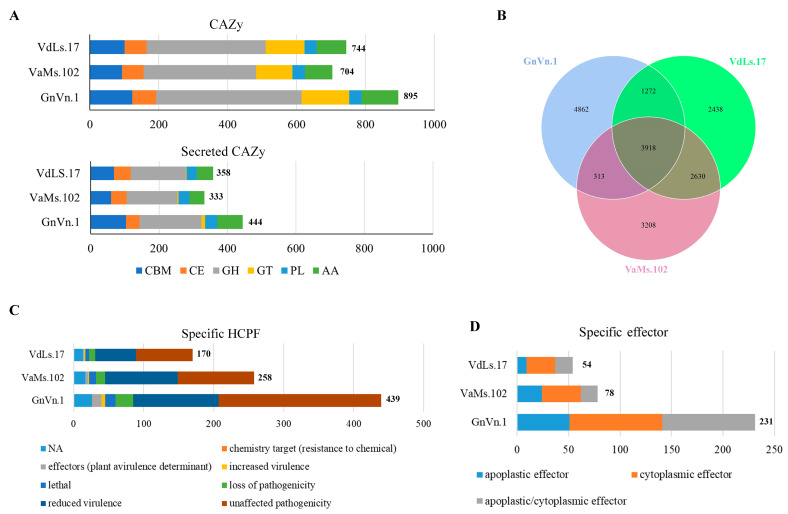
Core–pan analysis results of GnVn.1, VdLS.17, and VaMs.102. (**A**) Number of CAZys and secreted CAZys in GnVn.1, VdLS.17, and VaMs.102. CBM: carbohydrate-binding module, CE: carbohydrate esterase, GH: glycoside hydrolase, GT: glycosyl transferase, PL: polysaccharide lyase, AA: auxiliary activity. (**B**) Core–pan groups of GnVn.1, VdLS.17, and VaMs.102. (**C**) Number of specific HCPF genes in GnVn.1, VdLS.17, and VaMs.102. (**D**) Specific effector genes in GnVn.1, VdLS.17, and VaMs.102. The label and column length represent the number of genes.

**Table 1 jof-10-00838-t001:** Genome characteristics and assembly features of *G. nigrescens* GnVn.1.

Assembly Feature	*G. nigrescens* Vn-1
Total length of contigs (bp)	33,331,644
Genome size (Mb)	31.79
Number of contigs	22
Max contig length/size (bp/Mb)	6,280,609/5.99
N50 contig length/size (bp/Mb)	2,979,356/2.84
GC (%)	57.56
Complete and single BUSCOs (%)	98.3%
Number of predicted genes	10,876
Gene density (/Mb)	342
Gene total length (bp)	16,243,094
Average gene length (bp)	1493
tRNA	188
Repeats	640
Total length/percentage of repeats (bp/%)	79,953/0.24%

## Data Availability

All data are available in the Section 3 and Appendix A. The genomic data of GnVn.1 are available in the NCBI database at https://www.ncbi.nlm.nih.gov/datasets/genome/GCA_043790115.1/ (accessed on 1 December 2024) under accession number SAMN43317469. The sequence read archive data are available under accession number SRR31549079.

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
