# Peer review of "Genome Sequencing and Comparative Genomic Analysis of Attenuated Strain Gibellulopsis nigrescens GnVn.1 Causing Mild Wilt in Sunflower"

_jof, 2024, doi:10.3390/jof10120838_

Round 1
Reviewer 1 Report
Comments:
Abstract - More information regarding how the comparative genomic analysis helps identify specific virulence genes would be beneficial, even though the abstract is well-structured. The impact of the abstract might be enhanced with an extra phrase outlining the potential applications of the findings (such as biocontrol strategies).
Introduction - Although the background information in the introduction is useful, it would be beneficial to make clear the precise research need that was previously addressed. You say, for example, that there was no genetic data for Gibellulopsis nigrescens, but this should have been stressed earlier to make the problem description more precise.
The practical consequences of comprehending G. nigrescens' virulence and cross-protection for agricultural disease management should be more clearly connected.
Methods - It's excellent that the process is thorough. Nonetheless, readability might be enhanced by a graphic representation of the procedure (such as genome sequencing, gene prediction, and comparative analysis).
A more thorough explanation of the tools selected, especially in comparative genomic analysis, would improve this section. How do the pathogenic processes of these particular species connect to those of G. nigrescens, and why were they chosen for comparison?
Results - Although the results are comprehensive, the statistics require more readable titles and labels in order to be self-contained. Make sure that each figure can be understood without consulting the text.
To bolster the importance of G. nigrescens in terms of its capacity for biocontrol, you might wish to add more strain-to-strain comparisons. For clarity, several of the descriptions may be clarified, particularly those pertaining to particular genes linked to pathogenicity and their roles.
Discussion - Although the discussion does a decent job of connecting the findings to earlier research, some assertions might need more specific evidence to back them up. For example, additional information about how G. nigrescens' genome contributes to its low virulence could strengthen the claim that it is a viable biocontrol resource.
The debate would be strengthened by adding a section on limits and potential future study areas. In what ways do these results open the door to further genetic research or real-world applications?
Conclusion - The key findings should be briefly summarized in the conclusion, although it might be more forceful when talking about the research's wider implications. What possible ramifications can this study have for crop disease management and agriculture, for example?
refer major comments section
Author Response
Comments 1: Abstract - More information regarding how the comparative genomic analysis helps identify specific virulence genes would be beneficial, even though the abstract is well-structured. The impact of the abstract might be enhanced with an extra phrase outlining the potential applications of the findings (such as biocontrol strategies).
Response 1: Thank you for pointing it out. I agree with this comment. Therefore, I have added a sentence in the abstract (page 1, line26, 28-29). The change was highlighted in the manuscript.
Comments 2: Introduction - Although the background information in the introduction is useful, it would be beneficial to make clear the precise research need that was previously addressed. You say, for example, that there was no genetic data for Gibellulopsis nigrescens, but this should have been stressed earlier to make the problem description more precise.
Response 2: A good suggestion. I agree this comment. The sentence “Al Although it was discovered a long time ago, there are no available data on its genome” was addressed in the first paragraph in section of introduction (page 1, paragraph 1, line36).
Comment 3: The practical consequences of comprehending G. nigrescens' virulence and cross-protection for agricultural disease management should be more clearly connected.
Response 3: Thank you for pointing this out. I have adjusted the structure of first and second paragraph of introduction (page 1, paragraph 1, line 37-43). I recount that G. nigrescens is a pathogen at first paragraph, followed by it was attenuated pathogen. In this way, the attenuated virulence and cross protection were connected clearly.
Comment 4: Methods - It's excellent that the process is thorough. Nonetheless, readability might be enhanced by a graphic representation of the procedure (such as genome sequencing, gene prediction, and comparative analysis).
Response 4: A good idea. I have added a Figure in the manuscript (Figure 1) (page 4, line 149-150).
Comment 5: A more thorough explanation of the tools selected, especially in comparative genomic analysis, would improve this section. How do the pathogenic processes of these particular species connect to those of G. nigrescens, and why were they chosen for comparison?
Response 5: The reason why I chose Verticillium dahliae VdLs.17 and Verticillium alfalfa VaMs.102 for comparison and chose Fusarium oxysporum Fo47 and F. solani Fusso1 as an outgroup as outgroup of phylogenic tree, was added in the manuscript with highlight formatting (page 3, line 128-131; page 3 line 143-146).
Comment 6: Results - Although the results are comprehensive, the statistics require more readable titles and labels in order to be self-contained. Make sure that each figure can be understood without consulting the text.
Response 6: We have readjusted the figures, and added the labels. (result section, Figure 4-6)
Comment 7: To bolster the importance of G. nigrescens in terms of its capacity for biocontrol, you might wish to add more strain-to-strain comparisons. For clarity, several of the descriptions may be clarified, particularly those pertaining to particular genes linked to pathogenicity and their roles.
Response 7: A good suggestion. We have added more comparisons between Gn.Vn-1 and two strains in Verticillium genus. The changes were showed with highlighting formatting (result section, page 8, line 283-285 and line 287-295). We also added table S9 in supplementary files.
Comment 8: Discussion - Although the discussion does a decent job of connecting the findings to earlier research, some assertions might need more specific evidence to back them up. For example, additional information about how G. nigrescens' genome contributes to its low virulence could strengthen the claim that it is a viable biocontrol resource.
Response 8: A good suggestion! We have added discussion about how G. nigrescens' genome contributes to its low virulence. The detail was highlighted in manuscript (page 11, line, 373-383).
Comment 9: The debate would be strengthened by adding a section on limits and potential future study areas. In what ways do these results open the door to further genetic research or real-world applications?
Response 9: A good suggestion. This section is a guidance for further genetic research and attenuated strains application. The section was added in the manuscript with highlighting formatting (discussion section, page 11, line 392-398).
Comment 10: Conclusion - The key findings should be briefly summarized in the conclusion, although it might be more forceful when talking about the research's wider implications. What possible ramifications can this study have for crop disease management and agriculture, for example?
Response 10: We have simplified the key findings, and added the possible ramifications for crop disease management in the future. The change was highlighted in the manuscript (conclusion section, page 11, line 400-410).

Reviewer 2 Report
Review on “Genome sequencing and comparative genomic analysis of attenuated strain Gibellulopsis nigrescens Vn-1 causing mild wilt on sunflower” for manuscript ID jof-3288407
In brief introduction the authors highlight the significance of Gibellulopsis nigrescens as a biocontrol agent against plant pathogens, particularly its role in managing Verticillium wilt in crops. The introduction identifies a gap in the existing literature regarding the genomic characteristics of G. nigrescens, particularly the strain Vn-1, which has not been extensively studied.
Unfortunately, some recent studies of authors have been missed. The following paper could help to improve the Intro:
· Alisaac, E., Götz, M. First report of Gibellulopsis nigrescens on peppermint in Germany. J Plant Dis Prot 129, 207–209 (2022). https://doi.org/10.1007/s41348-021-00540-0
Major points:
There are no supplementary files in the submission, but many references to them along a manuscript: L204, L214, L246, L368 and so on.
Augustus 2.7 is rather old software (released 10+ years ago), the details of assembly aren’t clear. “PacBio Sequel” is not a assembly software.
L98, L148: BUSCO version and used database is required
Raw data need to be publicly available to reproduce authors’ results.
L154: The BioProject is empty, no publicly available data. No assembled genome is released to GenBank. The given assemblies (GCF_000150675.1 and GCF_000150825.1) are not the authors’ results and might be omitted in Data Availability section.
Instead of review about plant immunity [37] please provide the recent study of plant and pathogenic fungi interaction.
Legends of Fig 4 and Fig 5 lack details, Fig 5A should use the same scale.
Minor points:
L20: incorrect abbreviation, PHI stands for Plant-Host Interaction, please cite the PHI-base also https://doi.org/10.1093/nar/gkab1037
L33: “infected” → “infect”
L43: please rephrase the sentence, who controls what?
L94: “BLASR” → “BLAST”
L119: replace the [28] with current reference about AntiSMASH
Alisaac, E., Götz, M. First report of Gibellulopsis nigrescens on peppermint in Germany. J Plant Dis Prot 129, 207–209 (2022). https://doi.org/10.1007/s41348-021-00540-0
L106: please specify the version and used database of RepeatMasker.
Abbreviation of the strain name to Gn.Vn-1 is not common practice, please consider to specify full organism name along the whole manuscript.
Author Response
Comment 1: Unfortunately, some recent studies of authors have been missed. The following paper could help to improve the Intro: Alisaac, E., Götz, M. First report of Gibellulopsis nigrescens on peppermint in Germany. J Plant Dis Prot 129, 207–209 (2022). https://doi.org/10.1007/s41348-021-00540-0
Response 1: we have added this paper as reference in our manuscript. The order of reference is [8] (page12, line445).
Comment 2: There are no supplementary files in the submission, but many references to them along a manuscript: L204, L214, L246, L368 and so on.
Response 2: We have upload supplementary files in submission system. (supplementary files Table S1-S9)
Comment 3: Augustus 2.7 is rather old software (released 10+ years ago), the details of assembly aren’t clear. “PacBio Sequel” is not an assembly software.
Response: We have deleted “PacBio Sequel”.
Comment 4: L98, L148: BUSCO version and used database is required
Raw data need to be publicly available to reproduce authors’ results.
Response 4: BUSCO version and used database were added (page 3, line 100-101).
Comment 5: L154: The BioProject is empty, no publicly available data. No assembled genome is released to GenBank. The given assemblies (GCF_000150675.1 and GCF_000150825.1) are not the authors’ results and might be omitted in Data Availability section.
Response: The genome was not released, when I submitted the manuscript first time. I have rewrite the website and accession number of Gn.Vn-1 in Results and Data Availability Statement sections. The changes were showed in highlighting format (page 4, line 163; page 12, line 426-427).
Comment 6: Instead of review about plant immunity [37] please provide the recent study of plant and pathogenic fungi interaction.
Response 6: I have instead the reference [37]. The new reference order is [39]. (page 13, line 502)
Comment 7: Legends of Fig 4 and Fig 5 lack details, Fig 5A should use the same scale.
Response 7: A good suggestion! The figures and legends were modified. The figure order of revised manuscript was figure 5 and Figure 6. The figure legends were highlighted (Figure 5, page 8, line 268-270; figure 6, line 295-301)
Minor points:
Comment 8: L20: incorrect abbreviation, PHI stands for Plant-Host Interaction, please cite the PHI-base also https://doi.org/10.1093/nar/gkab1037
Response 8: I have revised the abbreviation to HCPF in the whole manuscript. The website was cited
Comment 9: L33: “infected” → “infect”
Response 9: The word has been corrected (page 1, line 40).
Comment 10: L43: please rephrase the sentence, who controls what?
Response 10: The sentence has been rephrased (page 2, line 47-48).
Comment 11: L94: “BLASR” → “BLAST”
Response 11: It has been corrected (page 2, line 97).
Comment 12: L119: replace the [28] with current reference about AntiSMASH
Response 12: Thank you for suggestion, the reference has been replaced, the new reference order was [29]. (page 13, line 483-485).
Comment 13: L106: please specify the version and used database of RepeatMasker.
Response 13: Specify the version and used database were added (page 3, line 108-109).
Comments14 : Abbreviation of the strain name to Gn.Vn-1 is not common practice, please consider to specify full organism name along the whole manuscript.
Response 14: It is inconvenient to use the full organism name in the whole manuscript, especially in the figures. We have changed the Gn.Vn-1 to GnVn.1 in the whole manuscript.
Round 2
Reviewer 2 Report
I would like to thank the authors for the improving the manuscript, but some concerns remain to be addressed.
Thanks for adding the genome assembly, but Bioproject is still lacks raw sequencing data (https://www.ncbi.nlm.nih.gov/bioproject/PRJNA1151617)
L101: reference to BUSCO required (f.e. https://doi.org/10.1002/cpz1.323)
Figure 1: BUSCO evaluation step repeats three times.
The question about using old version of Augustus remained unanswered.
L12: the phone numbers could be presented in international form
L322: [40.42] → [40, 42]
Author Response
Comment 1:I would like to thank the authors for the improving the manuscript, but some concerns remain to be addressed. Thanks for adding the genome assembly, but Bioproject is still lacks raw sequencing data (https://www.ncbi.nlm.nih.gov/bioproject/PRJNA1151617)
Response 1: Thanks for your suggestion, we have submitted the raw data on the NCBI Sequence Read Archive (SRA) database, with accession number of SRR31549079. The information was described in “Data Availability Statement” (page 12, line 429).

Comment 2: L101: reference to BUSCO required (f.e. https://doi.org/10.1002/cpz1.323)
Response 2: we have added the reference to BUSCO, the order of this reference is [24]. (page3, line101)
Comment 3: Figure 1: BUSCO evaluation step repeats three times.
Response 3: thanks for your question, we have corrected the figure1 (page 4, line 149)
Comment 4: The question about using old version of Augustus remained unanswered.
Response 4: Even though Augustus is an old tool, we discover many recent articles employ it for the ab initio prediction. We also used it for gene prediction. In addition, we used another tool GeneWise 2.4.1 for homology-based prediction for result validation.
Comment 5: L12: the phone numbers could be presented in international form.
Response 5: Thanks for your question, we have corrected the phone numbers. (page 1, line12-13)
Comment 6: L322: [40.42] → [40, 42]
Response 6: Corrected. (page10, line 321)
